# How to Predict Metastasis in Luminal Breast Cancer? Current Solutions and Future Prospects

**DOI:** 10.3390/ijms21218415

**Published:** 2020-11-09

**Authors:** Sylwia Tabor, Małgorzata Szostakowska-Rodzos, Anna Fabisiewicz, Ewa A. Grzybowska

**Affiliations:** Maria Sklodowska-Curie National Research Institute of Oncology, Roentgena 5, 02-781 Warsaw, Poland; Sylwia.Tabor@pib-nio.pl (S.T.); malgorzata.szostakowska@pib-nio.pl (M.S.-R.); anna.fabisiewicz@pib-nio.pl (A.F.)

**Keywords:** breast cancer metastasis, ER-positive, dormancy, hormonal crosstalk, multigene tests, circulating tumor markers

## Abstract

Breast cancer metastasis is the main cause of breast cancer mortality. Luminal breast cancer represents the majority of breast cancer cases and, despite relatively good prognosis, its heterogeneity creates problems with a proper stratification of patients and correct identification of the group with a high risk of metastatic relapse. Current prognostic tools are based on the analysis of the primary tumor and, despite their undisputed power of prediction, they might be insufficient to foresee the relapse in an accurate and precise manner, especially if the relapse occurs after a long period of dormancy, which is very common in luminal breast cancer. New approaches tend to rely on body fluid analyses, which have the advantage of being non-invasive and versatile and may be repeated and used for monitoring the disease in the long run. In this review we describe the current, newly-developed, and only-just-discovered methods which are or may become useful in the assessment of the probability of the relapse.

## 1. Introduction

Breast cancer (BC) is the most frequently-diagnosed cancer and the second leading cause of cancer-related death in women. Despite the improvement of early diagnosis and a growing progress in treatment strategies, metastatic breast cancer (MBC) remains incurable.

Breast cancer has high morphological and molecular heterogeneity not only between tumors but also within a single tumor. Gene expression profiling enables defining and characterizing of main intrinsic molecular subtypes [1,2] with different clinical behaviors and responses to treatment. The distinction between subtypes is based on concurrent expression of three tumor markers—estrogen and progesterone hormone receptors (HR) and HER2 (human epidermal growth factor receptor 2). Luminal A tumors are cancers with the highest expression of HR, they are HER2-negative and show low expression of proliferation-related genes. This subtype is the most diverse, tends to grow slowly and has a good prognosis. Luminal B subtype shows lower expression of HR and is either HER2-positive or -negative. This group shows higher levels of Ki-67 proliferation index, therefore grows faster and has worse a prognosis than luminal A subtype. HER2-enriched subtype comprises HER2-positive but HR-negative tumors which grow faster than luminal cancers and may have a worse prognosis. Triple-negative breast cancer (TNBC), which is hormone-receptor-negative and HER2-negative is itself heterogeneous, comprising different subtypes, including basal-like, immune-modulatory, luminal androgen receptor, mesenchymal, and claudin-low.

The highest survival rate is observed among women with HR+/HER2- subtype, followed by the HR+/HER2+ subtype, and the HR-/HER2+ subtype. The TNBC subtype has the worst survival rate. According to data from the Surveillance, Epidemiology, and End Results Program (SEER) program the 5-year relative survival rate in US between 2013 and 2017 was 90.4% for HR+/HER2- subtype, 83.6% for HR+/HER2+, and only 76.7% for TNBC subtype [3].

Luminal BC has relatively good prognosis, but since it constitutes about >70% of cases, metastatic relapse in luminal BC is in sheer numbers larger than in the other subtypes, especially in the long run. For example, in the study of Maaren et al. [4], which presented ten-year recurrence rates in different BC subtypes, the number of patients with distant metastases in luminal subtypes reached around 10%, while in other subtypes reached only 4% of all patients in the study, despite a significantly-lower recurrence rate in luminal subtypes. Thus, a proper classification of luminal BC tumors into metastasis-prone and unlikely-to-metastasize is considered a key issue in BC research, since it helps to determine the type of treatment. Treatment decisions are usually taken on the basis of various clinicopathological parameters including—besides the receptor status—tumor size and grade, lymphovascular invasion, and nodal and proliferation status. Currently, additional information provided by molecular profiling and liquid biopsy has become an integral part of these treatment decisions. This review will focus on describing how well can we predict the outcome using the current knowledge of tumor biology and the available tools and the challenges that still lie ahead.

## 2. The Role of HR Receptors and Hormonal Crosstalk in the Prediction of the Relapse 

Hormone-receptor status is the most important factor in breast cancer patients’ stratification. Combined estrogen receptor (ER) and progesterone receptor (PR) positivity is a good prognostic factor, observed in luminal A subtype, which has the best prognosis. However, it is important to properly perceive ER/PR interplay. While there is a general consensus about the role of estrogen, which is a known driver of tumor growth, but an inhibitor of breast cancer cell invasion [5,6], the role of progesterone is more complex. Progesterone is considered to counteract estrogen proliferative functions by competing with the same targets on chromatin (85% common targets) [7]. In isolation, estrogen and progestin act agonistically, regulating common targets. However, when both hormones are present, progestin behaves as an estrogen antagonist, activating the progesterone receptor, which remodels nucleosomes and redirects binding to chromatin by the estrogen receptor (Figure 1A) [7].

The estrogen receptor has two forms, α and β, and while the ER-α is an established driver of mammary tumors because it enhances proliferation, ER-β has been described as anti-proliferative and pro-apoptotic, thus acting antagonistically to ER-α. Its activity is however still disputed since some proliferative actions of ER-β have been described and the possibility of its bi-faceted role in BC has been suggested [8]. It was also observed that the level of ER-β in BC patients may help predict tamoxifen and anastrozole responsiveness in the neoadjuvant setting and that the ER-α/ER-β ratio can be predictive in BC [9].

One of the main survival-affecting events in breast cancer is the loss of the estrogen receptor positivity, resulting in resistance to ER-targeted therapies and eventually, progression to metastatic disease [10,11]. Interestingly, luminal BC metastases have been described as heterogeneous, containing ER+ and ER- lesions [12], indicating heterogeneous seeding or the selective loss of ER positivity in disseminated cells. Conversely, the discordance in PR status between primary tumor and metastatic lesions has not been reported as important for survival [10].

There are discrepancies in the accumulated knowledge about the role and predictive value of the progesterone receptor. The PR has been described as an independent predictor of response to tamoxifen [13] and its expression as correlating with decreased metastatic events in early BC [14,15]. However, new evidence suggests that the PR may be prognostic, but not predictive [16], and its cumulative impact represents an outcome of a complicated interplay. Due to its ability to redirect ER-mediated transcription, in addition to its own independent positive prognostic value, PR affects the prognostic value of the ER in ER+/PR+ cancers [7]. However, there are reports associating PR expression with increased invasiveness and metastasis [17,18].

As with the estrogen receptor (forms α and β), there are two forms of progesterone receptor, A and B, each with a different set of functions (there is also form C, but it lacks a DNA-binding domain, so it is not functional in terms of activating transcription). Overexpression of the progesterone receptor A (PR-A) has been reported in oncogenesis [19] and is associated with increased invasiveness [17] and worse overall survival [20]. PR-B also has been associated with increased invasiveness [21,22], but, as pointed out by MacFall et al. [17], these observations were made at high, luteal progesterone concentrations and without estrogen signaling. However, PR-B has also been implicated in c-Src/p21ras/Erk signaling, in which proliferation is promoted by progestins via ligand-activated PR-B, in cooperation with ER signaling, underlining the role of the ER/PR crosstalk [23]. Despite a documented role of progesterone in the inhibition of estrogen-induced proliferation, there are studies showing that progesterone may boost proliferation [24,25] and induce mammary stem cells [26] and that progestin treatment of breast cancer cells pretreated with estrogen shifted cell phenotype from luminal A to basal [27].

Clinical data do not always conform to in-vitro-predicted outcomes, underlining the importance of the hormonal crosstalk and the ratio of the steroid hormones and hormone receptors in patient at the given time. While hormonal crosstalk in luminal breast cancer is extremely important, quite surprisingly it is much debated and far from clear, probably due to the too simplistic experimental approaches, which often highlight only one aspect of the complicated pathway (single receptor events). The amount of each hormone and the progesterone/estrogen (P/E) ratio changes with age and the phase of the menstrual cycle; in pre-menopausal women the P/E ratio achieves around 10–20 in the follicular and >100 in the luteal phase, thus the progesterone is always in excess; in post-menopausal women the ratio is lower (around 2–10), so there is less progesterone, but the overall levels of both hormones are also much lower (Figure 1B). It is hard to assess the net result of these numbers, not only because the details about the crosstalk between these hormones are still to be established, but also because they may not translate directly into the pattern of transcription due to hormone availability (related to the amount of hormone-binding protein in serum), the expression levels of ER and PR receptors, the action of additional estrogens (estrone and estriol), and the crosstalk with the other receptors (androgen receptor and glucocorticoid receptor).

To illustrate the controversy, Mohammed et al. [28] suggested that progestins should be combined with anti-estrogen therapies, due to their anti-proliferative effect, but Singhal et al. [7] recommended a treatment with anti-progestins. Some insight may be gained from the studies analyzing the effects of hormone-replacement therapy (HRT) in postmenopausal woman. It has been demonstrated that combined HRT (including estrogens and progestin) is associated with increased risk of invasive breast cancer, compared to the group treated with estrogen monotherapy or placebo [29].

These data, together with the results pointing to the increased invasiveness caused by PR action [18] suggest that progestin should be used with caution. However, the interpretation also remains controversial, since the synthetic progestin (medroxyprogesterone acetate, MPA) used in HRT is long-lived compared to the natural ones [30], and this may affect the duration and exposure levels, as well as the crosstalk with estrogen regulation in patients.

## 3. Molecular Subtyping and the Current Molecular Diagnostic Tests

Hormone receptor status, crucial for qualifying the patient for endocrine therapy is routinely established by immunohistochemical (IHC) staining, but molecular classification, described by Perou [1] and Sorlie [2], allows for more accurate subtyping and provides more information. The growing database of molecular data allows sub-stratification of the four initial intrinsic subtypes into more specific categories, and an identification of additional molecular subtypes, with a different predictive and prognostic significance. While more costly and time-consuming than the standard IHC tests, these data are also more complete and useful in therapy choices.

Adjuvant endocrine therapy is a method of choice for almost all luminal breast cancer patients. Qualification for adjuvant therapy in luminal HER2-negative early breast cancer (stages I and II in the 8th edition of the American Joint Cancer Committee (AJCC) TNM classification for breast cancer [31]) remains one of the major clinical issues. Addition of systemic treatment to surgery and radiotherapy aims to achieve better locoregional control and suppress eventual micromestastases. Its type and duration depend on multiple factors such as tumor size, lymph node status, histologic grade, menopausal status, age, performance status, comorbidities, and the patient’s preferences. In premenopausal patients it consists of ovarian-function blockage and tamoxifen or an aromatase inhibitor (AI). For postmenopausal patients, an AI alone or received in some point of therapy is an option. According to the St. Gallen 2019 consensus five years of endocrine therapy is recommended both for groups in stage I and for postmenopausal patients in stage II with no lymph node involvement who have taken an AI for five years [32]. General indications for ten years extended endocrine therapy are stage II and lymph node involvement. The next method of systemic adjuvant treatment is chemotherapy which is highly recommended for patients with 4–9 positive lymph nodes apart from other pathological features [32]. Qualification for adjuvant chemotherapy followed by endocrine therapy or endocrine therapy alone in all of the remaining clinical advancement stages of early breast cancer is challenging. To overcome this problem, large-scale genomic studies have enabled the selection, without any bias or preexisting conception, a set of genes with a differential expression profile, which can be used to predict a risk of recurrence in BC. Therefore, over the last few years we have observed the development of many genomic tools which estimate a risk of recurrence based on expression of cancer related genes. The most-known of these tests are described below (listed in Table 1).

### 3.1. OncotypeDx Recurrence Score (RS)

OncotypeDx is a 21-gene assay designed to predict a risk of recurrence in early-stage, axillary lymph-node-negative luminal HER2-negative breast cancer. At times, when a clinical decision is highly complex, it can be used to predict a benefit of adding an adjuvant chemotherapy in postmenopausal, lymph node-positive breast cancer patients. OncotypeDx is the only multigene test included in 8th AJCC TNM classification [31]. It is also recommended by multiple of oncology associations like the American Society of Clinical Oncology (ASCO) [33] or the European Society for Medical Oncology (ESMO) [34] and incorporated in the National Comprehensive Cancer Network (NCCN) guidelines for breast cancer [35] and the St.Gallen consensus [32]. The assay is based on a reverse-transcriptase polymerase-chain-reaction (RT-PCR) of 21 genes performed on RNA extracted from a fragment of the formalin-fixed, paraffin-embedded tumor tissue (FFPE). Subsequently, the results from RT-PCR are calculated and showed in a score scale from 0 to 100. In the prospective TAILORx trial (the Trial Assigning Individualized Options for Treatment) patients were divided into three risk groups based on obtained score. Firstly, patients with a score from 0 to 10 who were considered as those with favorable prognosis and who were qualified to receive endocrine therapy alone. Then, patients with a score of 11 to 25, described as those with midrange risk, who were randomly assigned to obtain endocrine therapy alone or with chemotherapy. Lastly, the third group of patients, with a score of 26 or higher, who were given adjuvant chemotherapy followed by endocrine therapy. The results of the trial confirmed a low risk of recurrence in a first group measured by 98.0% (95% CI, 97.1 to 98.6) 5-year overall survival (OS) and 98.7% (95% CI, 97.9 to 99.2) freedom from distant and locoregional regress of disease [36]. Moreover, the results were independent from clinicopathological features. A secondary analysis of the TAILORx trial showed a benefit from adjuvant chemotherapy added to endocrine therapy in a cohort with RS of 26 or higher [37]. In clinical practice patients are divided into two groups based on a score—patients with low risk of recurrence who benefit from endocrine therapy alone (RS 0–25) and those with high risk of recurrence (RS 26 or higher) who should be qualified for adjuvant chemotherapy. NCCN guidelines distinguish a third group (RS 26–30) where both of these therapeutic approaches can be considered [35].

The Magee equation (https://path.upmc.edu/onlineTools/mageeequations.html) is a tool applied in clinical practice to estimate OncotypeDx score. It uses six routinely-assessed pathological values to calculate a necessity of performing additional molecular testing. Obtained results seem to be consistent with this multigene assay [38] and are much easier to use in everyday practice.

IHC4 is a similar prognostic test which evaluates four typical immunohistochemistry markers (ER, PR, HER2, and Ki-67). In a large study with a cohort comprised of 1125 subjects two methods of molecular diagnostic were compared. The 21-gene Genomic Health recurrence score (GHI-RS, OncotypeDx) was analyzed together with the IHC4 score. The study was designed to determine if these two assays differ in terms of providing clinically-important data. The results suggested that information obtained in the IHC4 score can be similar to that in the GHI-RS [39]. On the contrary, results of a prospective trial comparing the breast cancer index (BCI) assay, OncotypeDX, and IHC4 made on tissue originated from the TransATAC study population definitely pointed at the BCI-linear model (BCI-L) as the only statistically-significant score for both early and late distant recurrence [40]. Therefore, IHC4 is not recommended for use by most of oncology organizations.

### 3.2. MammaPrint (MP)

MammaPrint is a 70-gene signature which uses microarray profiling provided on a fragment of fresh tissue or FFPE. Not surprisingly, the genes in the MammaPrint signature were found to correlate with the six hallmarks of cancer (apoptosis, insensitivity to growth and anti-growth signals, replicative potential, invasion, and angiogenesis) [41]. Information obtained from the assay concerns early luminal HER2-negative breast cancer patients with both negative and positive lymph node status (N0-3) and is binary. Results are divided into two groups—low and high risk of 5-year recurrence. MP assay was validated in multiple clinical trials from which MINDACT (Microarray in Node-Negative and 1 to 3 Positive Lymph Node Disease May Avoid Chemotherapy) seems to be the most important. That was the first prospective randomized trial evaluating usage of adjuvant chemotherapy in high genomic and clinical risk in comparison to resignation of it in a group with low genomic but high clinical risk. The study proved that approximate 46% patients who were candidates for adjuvant chemotherapy based on clinical factors probably did not require it [42]. Moreover, patients with low clinical risk did not benefit from chemotherapy even if they were qualified to group with high genomic risk [43]. MP is a signature recommended by the ESMO [34] and included in the St.Gallen consensus [32].

### 3.3. Breast Cancer Index (BCI)

The Breast Cancer Index is an assay based on RT-PCR performed on a fragment of FFPE and designed to evaluate potential tumor responsiveness to hormonal therapy. It analyses two biomarkers and the expression of five genes. Its prognostic value is described above. BCI is not recommended by most of oncology associations due to insufficient evidence.

### 3.4. Prosigna PAM 50 Risk of Recurrence Score (ROR)

Prosigna PAM 50 is an assay based on RT-PCR analysis of 50 genes made on a fragment of a tumor tissue. It predicts 10-year survival rate from clinicopathological parameters combined with gene profile results. Its prognostic value was proven as statistically-significant in luminal, early breast cancer patients who did not receive adjuvant chemotherapy [44]. Although it does not have any predictive value, patients in the low ROR category can be considered as optimal for exclusive endocrine therapy. ROR is endorsed in the St.Gallen consensus [32].

### 3.5. EndoPredict (EP)

EndoPredict is a 12-gene molecular assay which evaluates a risk score solely or in combination with nodal status and tumor diameter. The GEICAM 9906 trial has shown that it can be an independent prognostic factor for luminal, HER2-negative, lymph-node-positive early breast cancer [45]. The St. Gallen consensus [32] permits usage of EP if indicated.

### 3.6. Serum Biomarkers

In addition to primary tumor analysis, serum biomarkers—cancer antigen 15-3 (CA 15-3), carcinoembrionic antigen (CEA), and cancer antigen CA125 have been reported as elevated in metastatic breast cancer. Several studies investigated a correlation between markers’ expression and biological subtypes in BC. Yerushalmi et al. [46] and Li et al. [47] reported more significant elevation of CA 15-3 in luminal subtypes, while Fang et al. [48] observed that CA125 elevation was more significant in TNBC, which was, however, not confirmed in the study by Li et. al. [47]. Lian et al. [49] did not observe correlation of the markers with biological subtypes.

## 4. Luminal Tumor Biology and Its Implications on Metastasis

Molecular profiling helps greatly in risk assessment, but the biology of tumor progression still contains many unknown variables, which should be taken into account for more accurate prediction. These variables include the lack of a detailed knowledge about different routes of metastatic seeding, and if they correlate with different subtypes, the phenomenon of tumor dormancy and re-activation, tumor heterogeneity, and the constant evolution of tumor cells. All these factors challenge the accuracy of prediction, especially when it was established on the basis of primary tumor characteristics.

### 4.1. Models of Metastasis

As comprehensively described elsewhere [50,51,52] there are at least two main routes of breast cancer dissemination: (1) the classical route, termed linear progression, in which breast cancer cells of the primary tumor undergo a transition to more invasive phenotype, degrade the surrounding extracellular matrix, and actively intravasate and (2) the parallel progression route in which tumor cells may spread even before the primary tumor manifests clinically, just after the in situ lesion starts an angiogenic process (Box 1). While it seems logical that the classical route should be linked to epithelial–mesenchymal transition (EMT) and therefore should be expected rather in TNBC, and the parallel progression followed by dormancy should be more frequent in luminal breast cancer, the actual evidence is scarce. Early dissemination associated with the parallel progression model was mostly demonstrated in HER2-positive cancer [53,54], but it was also observed for ER-positive tumors [55]. Expanding this kind of research is vital since the association of the specific subtype with the particular metastatic route can determine if the therapeutic intervention should focus on preventing the spread or on maintaining the state of dormancy of the already-disseminated cells.

Box 1Linear progression—the two models of metastatic dissemination.(1) Linear progression–classical model of sequential events in which cells from the primary tumor gradually acquire migratory and invasive properties, usually linked to epithelial–mesenchymal transition (EMT). Occurs with degradation of the extracellular matrix, which enables tumor cells to actively intravastate. Requires time to accommodate phenotypical changes.(2) Parallel progression–dissemination occurs at an early after the pro-angiogenic switch, enabling tumor cells to intravasate viashedding into leaky vessels.

### 4.2. Dormancy in Luminal BC

Systemic minimal residual disease can be asymptomatic for years or decades due to a dormant, disseminated tumor cells (DTCs), which are temporarily subdued and quiescent, but may be induced, giving rise to a metastatic lesion. Dormant cells display growth arrest, which precludes proliferation (meaning that they are hard to eliminate with the standard cytotoxic treatment) and their survival mechanisms are strong. Thus, to effectively remove the threat of their arousal, two possible approaches can be taken—they may be eradicated against the odds via more direct, targeted approach, or they may be therapeutically kept in a dormant state. For both approaches more comprehensive understanding of DTC biology is needed.

In luminal breast cancers, dormancy is very pronounced and can be assessed by a specific signature for high dormancy scores (HDS), in contrast to triple-negative subtypes with low dormancy scores [56].

DTC presence can be established by the aspiration of bone marrow, but it is not a technique that can be used routinely in patients, since it is invasive and, due to the rarity of these events, negative results are not completely reliable and too much depends on a chance. The presence of DTCs can also be assessed by molecular analysis. Detailed knowledge pertaining to DTC biology should help in a detection of the existing dormancy and the assessment of the probability of relapse. The established ‘dormancy signature’ includes 22 up-regulated and 27 down-regulated factors [56], including NR2F1 (up-regulated). The expression of NR2F1 in DTCs from bone marrow aspiration samples has been proposed as a marker of dormancy [57]. It was demonstrated to play a part in the induction and the maintenance of dormancy by integrating epigenetic regulation of quiescence and survival [58].

MSK1 kinase has been also described as a regulator of metastatic dormancy in ER+ breast cancer; its low expression is associated with early relapse and increased bone homing. Assessing the expression of MSK1 has been proposed as useful in stratification of patients with ER+ breast cancer [59] in regard of the low or high risk of early relapse.

Another molecular indication of dormancy is associated with the ERKl/2/p38 ratio. As demonstrated by Sosa et al. [60], ERKlow/p38high ratio is observed in dormant cells leading to the three integrated events: (1) G0-G1 arrest, (2) mTOR activation which promotes basal survival, and (3) induced expression of the chaperone BiP/Grp78, which prevents apoptosis, but only in response to stress (adaptative survival). In contrast, ERKhigh/p38low ratio is observed in proliferating tumors and DTCs exiting dormancy [60].

The presence of DTCs has been associated with poorer prognosis [61], but only 40–60% of patients with DTCs in bone marrow will suffer from metastatic disease [62]. Thus, while the detection of dormancy may have some impact on prognosis, the main challenge lies in a correct assessment of the probability of DTC activation. Re-awakening of dormant cells involves various types of microenvironmental cues. Mechanisms proposed as the triggers are numerous and include hormonal signaling [12], angiogenic switch [63], immunosurveillance [63], NET formation (neutrophil extracellular traps, see Box 2) [64], interaction with extracellular matrix (ECM), and a crosstalk with stromal cells [65]. Some of these triggering events can be detected and quantified, which may have certain prognostic potential (see section ‘Circulating markers; established tests and future perspectives’).

Box 2Neutrophil extracellular traps (NETs).Neutrophil extracellular traps (NETs) are three-dimensional;
structures formed by decondensed chromatin, histones, DNA, and neutrophil granular proteins, like neutrophil elastase (NE), serine protease, and; myeloperoxidase (MPO). These web-like structures are released by activated neutrophiles in the NETosis process [66] in response to inflammatory stimuli. NETs were characterized, as a part of the innate immune response, as able to entrap and kill pathogens, but the newest research highlighted the role of NETosis in tumor microenvironment formation, tumor proliferation, tumor-associated thrombosis, and distant metastasis [64,67,68]. It has been demonstrated that NET-DNA; may act as a chemotactic factor that attracts cancer cells to distant locations. NET complexes that presented in the liver or lungs were found to; attract cancer cells to form distant metastases in some mouse models of breast cancer [69]. It was also shown that targeting NET in vivo reduces metastasis [68].

Tumor microenvironment (TME) is a powerful factor that plays a role in a termination of dormancy and metastatic relapse in general. TME consists of stromal cells and extracellular matrix (ECM), but it is also shaped by various immune cells (tumor-immune microenvironment, TIME). A crosstalk between tumor and stromal cells and an extracellular matrix (ECM)-mediated signaling are important for priming the pre-metastatic niche [70]. The factors involved in this priming include growth factors, cytokines, and exosomal factors and can be identified by a differential analysis of tumor cells secretomes [71].

Although molecular indicators of dormancy are promising and may prove their prognostic and predictive value in selecting patients susceptible to metastasis, they still have to make their route to the clinic.

### 4.3. Challenges of Tumor Heterogeneity and Clonal Evolution

Breast cancer is characterized with a high degree of inter- and intratumoral heterogeneity present on many levels including morphology (>20 histological types), genetic and molecular heterogeneity, and temporal phenotype conversion [72]. There are two main explanations of heterogeneity—the cancer stem-cell (CSC) hypothesis and the clonal evolution hypothesis. In the CSC hypothesis only a small fraction of tumor cells, with a stem-cell characteristic, is able to drive tumor progression [73]. In the clonal-evolution and selection model tumor cells constantly evolve, possibly in response to the internal selective pressure of tumor microenvironment or in response to the treatment [74].

Tumor heterogeneity poses impediments for therapeutic decisions—a small, undetected fraction of cancer cells may be responsible for the progression, but its genetic profile is unknown; also, metastatic lesions may differ in properties from primary tumor cells due to their origin from this undetected fraction or due to evolutionary changes and clonal selection. Thus, analysis of the primary tumor alone is not sufficient for the proper choice of treatment and monitoring of the disease. The analysis of circulating tumor markers described in the next section, may provide some solutions.

## 5. Circulating Markers; Established Tests and Future Perspectives

While available tests are able to predict metastasis in BC with some accuracy, for the reasons described in the previous section there is still a need to refine and improve these methods. Emerging new markers may contribute to this goal, especially because they are non-invasive, can be analyzed from easily-obtained body fluids (mostly blood samples) and provide information pertaining to the current status of the tumor. As depicted in Figure 2, most of the commonly-used methods rely on the analysis of the primary tumor sample, but this may shift towards a new, non-invasive analysis of circulating tumor markers, described below. The most advanced of these markers, validated for clinical use, are cell-free DNA (cfDNA) and CTCs.

Tumor cells may intravasate and disseminate early on, immediately after neoangiogenesis, as predicted by the parallel progression model, or as a consequence of a sequential, multi-step process (linear progression model). The crosstalk between the tumor cells and stromal and immune cells remodels extracellular matrix and creates populations of cancer-associated stromal and immune cells, facilitating intravasation. Primary site cells extrude signaling molecules and exosomes, which have a potential to prime other tissues for the subsequent dissemination, creating a pre-metastatic niche. Available and developed detection methods (marked in green) contribute to an assessment of the probability of metastasis.

### 5.1. cfDNA

The use of circulating cell-free DNA (cfDNA, Box 3) has shown its great potential in early detection, drug resistance, tumor relapse, and prediction of clinical outcomes in cancer patients. It has been shown in a few studies that cfDNA concentration is significantly higher in breast cancer patients than in healthy donors [75,76,77,78,79]. However, meta-analysis of 69 studies did not find cfDNA levels to be a valuable biomarker for early detection of BC [80]. Despite that, cfDNA levels were found to be a good biomarker for progression free survival (PFS), OS, and response to treatment [81]. As the cfDNA represents all circulating DNA, it is believed that analysis of ctDNA would be more informative. The ctDNA is identified in the cfDNA by detection of alterations characteristic to the tumor [82,83]. It has been shown that independently, high levels of ctDNA are poor prognostic factors for ER+ BC [84]. Moreover, changes in the genetic signature of ctDNA may reflect the changes in cancer cells that usually occur as a response to treatment, leading to resistance and/or metastases formation. The challenges in ctDNA monitoring are associated with the minimal amount of DNA as a template in genetic studies. Therefore, techniques used in the ctDNA-monitoring studies, include more sensitive platforms such as qPCR, ddPCR, Next Generation Sequencing (NGS), and whole-genome NGS (reviewed: [85]). For estrogen-positive BC it is crucial to detect genetic aberrations associated with resistance to anti-estrogen therapy, such as mutations in the ESR1 or mTOR/AKT-signaling pathway (reviewed: [11]). During the past few years, there have been multiple studies evaluating techniques and the clinical potential of ESR1 and PIK3CA mutations in prediction of resistance to an anti-estrogen therapy [84,86,87,88,89,90]. These studies are coherent about the clinical utility of detected mutations, however, there are plenty of discrepancies in mutation frequency and detection techniques. Therefore, before it is possible to rely on ctDNA analysis, a unification of the detection techniques and clinical validation of sensitivity, false positive result ratio (FPR), and false negative results ratio (FNR) must be achieved.

Box 3Circulating Tumor Markers.
**Circulating tumor DNA (ctDNA) and circulating cell-free DNA (cfDNA)**
cfDNA is released by apoptotic or necrotic cells, that
release nucleic acids into circulation. ctDNA is known to be a specific type of cfDNA that is secreted by living or apoptotic cancer cells. Both circulating nucleic acids may be detected in the bloodstream of the cancer patients. The alterations detected in ctDNA include aberrant mutations, hypermethylation, and copy number variations (reviewed: [82]).
**Circulating Tumor Cells (CTCs)**
CTCs are tumor cells circulating in the blood of patients
with solid tumors which may extravasate and form a metastatic lesion. They are
observed as single cells, CTC clusters, or conglomerates of tumor, stromal,
and immune cells. While single CTCs are very rare, CTC clusters are even less
common, but have higher metastatic potential [91,92]. There are various methods of CTC detection, but the most popular are based on the isolation of EpCAM (epithelial cell adhesion molecule)-positive cells.
**Circulating miRNAs**
MicroRNAs (miRNAs) are short, single-stranded RNAs, that are responsible for post-transcriptional gene expression regulation [93]. miRNAs are frequently deregulated in cancer and may act as oncogenes and/or tumor suppressors [94].

**Circulating extracellular vesicles**
Extracellular vesicles (EVs) are produced by most eukaryotic cells through an endosomal pathway and circulate in body fluids [95]. EVs comprise many vesicle types, of which the most studied are exosomes—plasma-membrane-enclosed particles of about 0.03–2 µm. Exosomes derived from cancer cells contain proteins and nucleic acids (miRNAs, lncRNAs, mRNAs, DNA, and mtDNA) representative to tumor tissue. Exosomes secreted from tumor cells can remodel the tumor environment by promoting metastasis and multidrug resistance (reviewed: [96]). Moreover, exosomes may be used as drug delivery vesicles for gene therapy [97,98].

### 5.2. Circulating Tumor Cells (CTCs)

For more than a century it has been known that cancer cells detach from the tumor lesion and circulate in the bloodstream, leading to metastasis [99], (Box 3). The estimation of the prognostic value of CTCs detected in the blood of breast cancer patients has recently been one of the most interesting topics in breast cancer research [100,101,102,103]. CTC presence is assumed to be prognostic for both luminal and triple-negative breast cancer (reviewed: [104]). Moreover, as CTCs are derived from both primary tumor and metastatic lesions, they may serve as representatives of the heterogeneous cancer cell populations. The number of CTCs in the blood of a patient was reported to be a statistically-significant factor in assessing the patient’s outcome. In breast cancer, CTCs are detected in about 20–30% of early and around 60% of advanced patients [105]. The advantage of CTC analysis over cfDNA is that we are able to analyze genetic, epigenetic, protein, and expression profiles of living, tumor-delivered cancer cells. Recent studies highlighted the importance of genetic variations in CTCs detected by NGS [106,107,108] and single-cell RNA-sequencing (scRNA-seq) [109]. CTCs represent excellent material for non-invasive cancer research, but they are very rare [104]. Currently, the only FDA-approved method for CTC identification, enumeration, and isolation is the CellSearch system (www.cellsearchctc.com). This system is Epithelial Cell Adhesion Molecule (EpCAM) dependent, which might be limiting, because tumor cells do not always express EpCAM, especially after epithelial–mesenchymal transition (EMT). Nevertheless, the usefulness of CTC detection has been validated in many clinical trials [52] and complements current prognostic models based on primary tumor characteristics and response to therapy [110].

### 5.3. Circulating miRNAs

Circulating miRNAs (Box 3) were found to be very stable in body fluids, including plasma, serum, urine, and breast milk (reviewed: [111]), therefore they have a great potential as a non-invasive biomarkers. Over the past decade there were multiple studies selecting most specific miRNAs for monitoring breast cancer progression, early recurrence, and response to treatment [112,113,114,115,116,117,118]. Recent reports showed that miRNAs may regulate processes of resistance to therapy, including anti-estrogen therapy (reviewed: [119]). Multiple studies about miRNAs expression patterns reported at least 96 up-regulated miRNAs and 87 down-regulated miRNAs in breast cancer tissues (reviewed: [120]). However, there are still no non-invasive tests using miRNA’s expression signature. The challenges in developing these kinds of tests comprise the lack of homogeneity between similar studies and inconsistencies in study parameters and methodology between studies. Moreover, large number of studies use in vitro models, which cannot be directly applied to clinical practice, as the cell culture conditions do not reflect the conditions of the human body. There is still lack of large-scale, patient-based clinical studies testing the utility of potential circulating miRNAs.

### 5.4. Exosomes

Exosomes (Box 3) have great potential as non-invasive biomarkers for cancer diagnosis, progression, and treatment response. Their content can be analyzed in genetic and proteomic analyzes. Exosomal non-coding RNAs are currently one of the most promising markers for breast cancer (role of non-coding RNAs in breast cancer reviewed in: [121]). Long non-coding RNAs are key epigenetic regulators of protein expression. Similarly to miRNAs, lncRNAs also can function as oncogenes and as tumor suppressors. Recent reports highlighted the role of lncRNAs in breast cancer progression (reviewed: [122]). Also, the exosomal miRNAs have been recently identified as one of the key regulators in metastatic breast cancer progression and treatment resistance (reviewed: [123,124]. Some new studies also highlighted the importance of proteomic analysis of exosomes. Mass spectrometry analyses of breast cancer cell-line-delivered exosomes and patient-derived scaffolds revealed that exosomes are a rich source of breast-cancer-related proteins and surface biomarkers that may be used for disease diagnosis and prognosis [125,126,127]. Exosomes are very promising non-invasive biomarkers not only because of the high spectrum of information they may deliver, but also because they can be found in all body fluids, including urine [128].

### 5.5. Neutrophil Extracellular Traps

As the evidence of the contribution of neutrophil extracellular traps NETs (Box 2) to the formation of distant metastases is growing, the ability to detect circulating NETs or NETosis levels in patients’ bloodstream may likely become a significant biomarker used to designate patients with high risk of metastatic progression. Current in vivo studies for detecting NETosis are based on the measurements of the NET-associated products, like circulating cell-free DNA, citH3, NE, and MPO in peripheral blood (reviewed: [129]). One of the most recent studies showed that serum levels of NET markers can predict the occurrence of liver metastases in patients with early-stage breast cancer [69]. Furthermore, it was found that the levels of NE–DNA complexes in serum of breast cancer patients correlate with the stage of the disease. Higher levels of NE–DNA complexes were observed in regional and metastatic disease [130]. Interestingly, NET formation was reported in a connection to a high-risk luminal A breast cancer subtype, with increased cell mobility, cancer-stem-cell-like features and hormonal signaling, but decreased proliferation and dysfunctional immune response [131].

## 6. Conclusions

Proper patient stratification and reliable discrimination between cases of high and low risk of metastatic relapse is of key importance in luminal breast cancer. While currently-available methods enable the prediction of the probability of metastasis with some accuracy, there are still many uncertainties, due to tumor heterogeneity and not-completely-understood mechanisms of metastatic dissemination. In luminal BC, factors like tumor cell dormancy and hormonal crosstalk are of particular importance, along with the more universal mechanisms, including ECM remodeling and stromal and immune cell involvement in tumor cell survival and dissemination. The other crucial and prediction-confounding factor is that tumor cells constantly evolve, acquiring new characteristics, for example estrogen resistance in the case of ER-positive BC. These new characteristics cannot be concluded from primary tumor samples. Thus, while most of the currently-used methods explore primary tumor tissue, newly-developed methods tend to be non-invasive and based on body fluid analysis. These emerging techniques are being validated for clinical use and offer a new perspective in monitoring luminal BC. Taking into account a number of studies actively researching best tools and biomarkers for BC non-invasive monitoring, it is possible that in few years there will be several new clinically-validated methods to support treatment decisions.

## Figures and Tables

**Figure 1 ijms-21-08415-f001:**
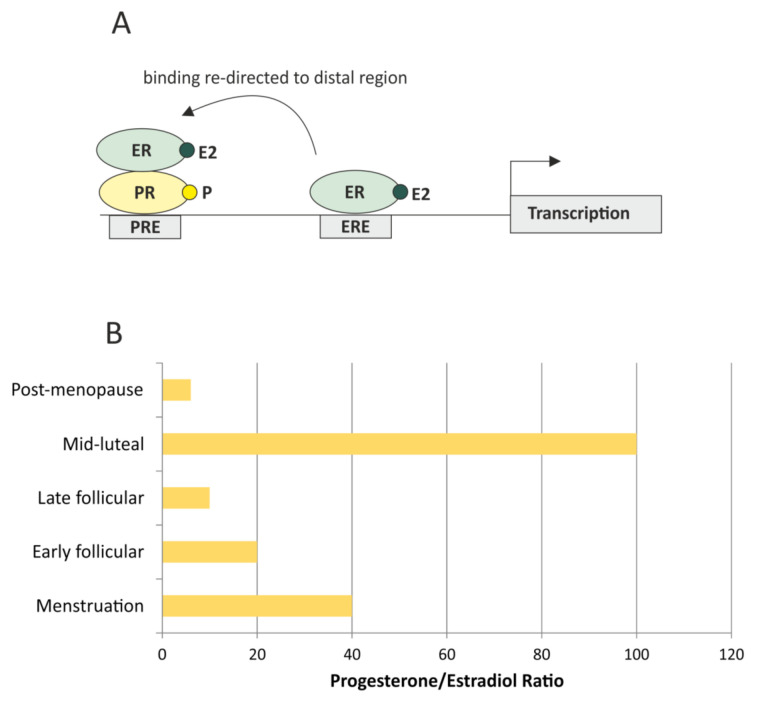
Estrogen receptor (ER) chromatin binding is re-programed in a presence of progesterone-bound progesterone receptor (PR). (**A**) In the presence of estradiol, ER binds to estrogen-responsive elements (ERE) in a proximal region of the promoter, but in a presence of both hormones, estradiol and progesterone, hormone-bound receptors form a complex which binds to more distal progesterone-responsive elements (PRE), re-programing the transcription profile. (**B**) The progesterone/estradiol ratio changes depending on the phase of menstrual cycle, assuming the lowest numbers in post-menopausal women (approximate ratio based on a data from The Global Library of Women’s Medicine). The ratio may influence the transcription profile due to a regulation described above.

**Figure 2 ijms-21-08415-f002:**
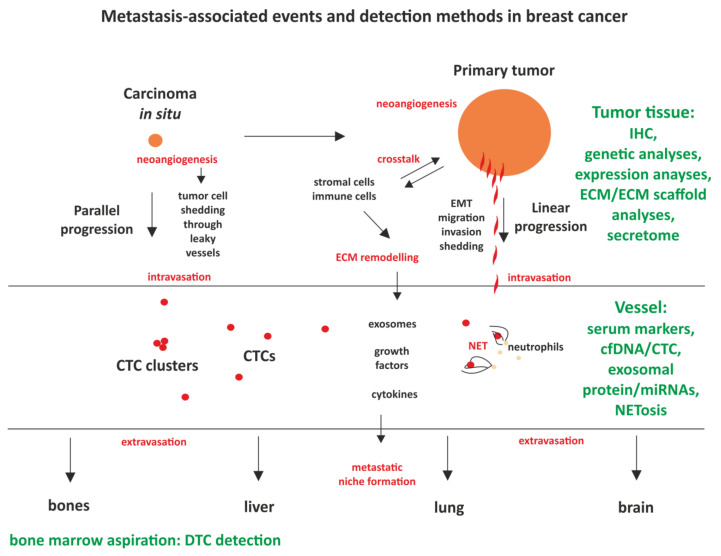
Metastasis-associated events and the methods of their detection in breast cancer patients.

**Table 1 ijms-21-08415-t001:** Genomic test designed to assess the risk of recurrence in BC.

Test	Number of Genes/Proteins/Parameters	Lymph Node Involvement	Prognostic/Predictive Value	Recommendations	References
OncotypeDx (Genomic Health, Inc.)	21 genes	N0	+/+	ASCO, ESMO, NCCN, St. Gallen	[32,33,34,35,36,37]
MAGEE (Agendia BV)	6 parameters	N0	+/+	-	[38]
IHC4	4 proteins	N0	+/+	-	[39,40]
MammaPrint	70 genes	N0-1	+/+	ESMO, St.Gallen	[32,34,41,42,43]
Breast Cancer Index (Bio Theranostics)	5 genes + 2 biomarkers	N0	+/+	-	[40]
Prosigna PAM 50 (Nanostring)	50 genes	N0	+/−	St.Gallen	[32,44]
EndoPredict (Sividon Diagnostics)	12 genes	N0-1	+/+	St. Gallen	[32,45]

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
