# Peer review of "How to Predict Metastasis in Luminal Breast Cancer? Current Solutions and Future Prospects"

_ijms, 2020, doi:10.3390/ijms21218415_

Round 1

Reviewer 1 Report

In the present review, Tabor et al. attempt to address the question of how to predict metastasis in luminal breast cancer.

The abstract and the conclusions sections are written in a more clear manner as compared to the main text. It becomes obvious from the abstract that the molecular profiling and downstream analysis of the primary tumor itself will most likely be insufficient to provide clinicians with the required information for the stratification of luminal breast cancer patients into patients with a metastasis-prone disease and patients with a more indolent disease (mainly due to the late relapse that is observed in the particular cancer type). The development of non-invasive tests (implying the liquid biopsy field) that could provide biopsy material in real-time for patient monitoring and treatment decision making is briefly mentioned.

Even if the research question is very interesting and the second part of the manuscript reads well, the review overall does not have a clear framing, it is not written in a way to support the main question and to guide the reader to some insightful conclusions. Instead, the main part seems like it consists of small independent reviews of different subtopics (models of metastasis, DTCs and dormancy, hormonal crosstalk, diagnostic tests and liquid biopsies) that do not have a coherent flow while reading them.

Some interesting points in the “future perspectives” section are mentioned; however, the section constitutes again a small review of liquid biopsies instead of suggesting sound future research directions deriving from the 1st part of the manuscript, which is very confusing.  

More specifically, after the generic part of the introduction (breast cancer subtyping into luminal A, luminal B, HER2+ and TNBC based on the molecular profiling) and the survival rate of each group and the purpose of the review, the models of the metastasis are mentioned, but only DTCs are described in the next section whereas CTCs, EVs etc will not be mentioned until much later in the future perspectives. The dormancy of DTCs is extensively described (even if dormancy applies to both DTCs and CTCs). The authors go back to the hormonal crosstalk and the suggested roles of HR and PR before moving to the current diagnostic tests and finally the future perspectives where circulating biomarkers (liquid biopsies) are described.

A suggestion would be to re-structure the main text (after introduction) as following:

Subtyping of breast cancer based on molecular profiling (phenotypic characterization)-> Existing knowledge on roles of progesterone and estrogen receptors, involved pathways and (ant)agonistic activities of estrogen and progestin (section 4) -> limitations of previous studies (eg study of the action of solely one hormone on specific pathways) and existing challenges (hormonal crosstalk) -> genotypic characterization of tumor cells for better discrimination of subtypes and current diagnostic tests (section 5) for a better understanding of the underlying biology and association with molecular profiling -> why analysis (phenotypic or/and genotypic) of primary tumor alone is not sufficient in treatment monitoring (heterogeneity, evolution, TME) and models of metastasis (part of section 2) with figure 2 and section 6 (explaining cfDNA, DTCs, CTCs etc) -> driving forces of seeding of tumor material (presence of heterogeneity, never- ending evolution) and knowledge on dormancy and re-activation regulators (TME etc). At that point, the main text should lead to interesting future perspectives (screen the patient with a more holistic approach).

Author Response

The reviewer points out certain important flaws in the logical structure of the manuscript and the general flow of narration. Complying to these helpful recommendations, we have re-structured the main text, and we believe that in this new form it is much more coherent, with the subsequent sections logically following each other. 

The structure of the revised version generally corresponds to the suggested order. The expanded introduction is followed by a section describing the role of hormone receptors and the hormonal crosstalk, their potential impact on metastasis and the existing challenges and the gaps in a current knowledge. Next, we describe molecular subtyping and the current molecular diagnostic tests, because these issues are closely related and correspond to each other. In the next section we analyze luminal tumor biology and the challenges posed by it, describing different possible models of metastatic seeding and dormancy after dissemination as potentially important factors in the future treatment decisions - providing that we would have the knowledge and tools to accurately assess and maintain dormancy. We described dormancy in disseminated cells, since – while it is of course possible to characterize dormancy signature in CTC, the lifetime of CTCs is generally measured in hours, so this phenomenon mostly pertains to DTC. Thus, we would like to describe it separately from the circulating markers - but the section pertaining to the circulating markers follows immediately after an explanatory chapter describing heterogeneity. We added the chapter about tumor heterogeneity in BC as an important part of tumor biology, explaining its origins and consequences, with the main conclusion that due to spatial and temporal heterogeneity, the analysis of primary tumor alone is insufficient for monitoring of the disease, and that circulating tumor markers may provide necessary information. This conclusion leads logically to a detailed description of those markers, how they are exploited today, if they are used in clinical practice and what are the future perspectives in body fluids analysis. 

We believe that these changes greatly improved the manuscript, and serve well its main purpose, which is to present in an easy to follow manner the current methods, proposed solutions and existing challenges in the prediction of metastatic relapse in luminal BC.  

Reviewer 2 Report

The work appears well set up and does not neglect the most well-known elements of study of luminal carcinoma.

1.The paper is a review, therefore certainly exhaustive and original, including all the methods concerning the topic.

2. I found it understandable and well written, but I certainly do not have the experience in the English language to certify its absolute correctness.

3. The conclusions are correct. I suggested, to improve its relevance, to identify among the emerging diagnostic methods which ones are in pool position to become routine diagnostics.

If it were possible I would emphasize the possibility that some of the tests mentioned could become a real diagnostic test of routine use.

Author Response

We very much appreciate the reviewer’s comments. The information about clinical utility of the described methods is included in the revised version.